# Vaccination during the First Diagnosis of Multiple Myeloma: A Cohort Study of the French National Health Insurance Database

**DOI:** 10.3390/vaccines8040722

**Published:** 2020-12-02

**Authors:** Guilhem Tournaire, Cécile Conte, Aurore Perrot, Maryse Lapeyre-Mester, Fabien Despas

**Affiliations:** 1Service de Pharmacologie Médicale et Clinique, CHU de Toulouse, 31000 Toulouse, France; guilhem.tournaire@yahoo.com (G.T.); cecile.conte@univ-tlse3.fr (C.C.); maryse.lapeyre-mestre@univ-tlse3.fr (M.L.-M.); 2UMR1027, Inserm, Université Paul Sabatier, 31330 Toulouse, France; 3Service de Pharmacologie Médicale et Clinique, Faculté de Médecine, Université Paul Sabatier, 31000 Toulouse, France; 4Centre Midi-Pyrénées de Pharmacovigilance, de Pharmacoépidémiologie et d’Informations sur le Médicament, Centre Hospitalier Universitaire de Toulouse, 31000 Toulouse, France; 5Département d’Hématologie et de médecine Interne, Institut Universitaire du Cancer-Oncopole, CHU de Toulouse, 31000 Toulouse, France; perrot.aurore@iuct-oncopole.fr; 6INSERM Centre d’Investigation Clinique 1436 Toulouse, Centre d’Investigation Clinique de Toulouse, Centre Hospitalier Universitaire de Toulouse, 31000 Toulouse, France

**Keywords:** pharmacoepidemiology, vaccines, French national health insurance database, multiple myeloma, infectious diseases

## Abstract

Purpose: Infections are frequent and often result in serious complications in patients with multiple myeloma (MM). Prophylactic vaccination is recommended for influenza virus, *Streptococcus pneumoniae (SP)*, and *Hemophilus influenzae*
*b (Hib)*. The aims of this study were to measure the vaccination rates within 24 months after the diagnosis of multiple myeloma and to identify factors associated with vaccine use. Methods: MM patients were selected through the French national health insurance database from 1 January 2010 to 31 December 2015. Patients with a previous history of MM were excluded. Results: Vaccination rates against influenza, *SP,* and *Hib* among 22,831 newly diagnosed MM patients were, respectively, 28.5%, 10.3%, and 1.4%. Only 0.7% received all three vaccines. Factors associated with vaccination were young age, male gender, an absence of comorbidity, a history of higher medication and vaccine consumption, Herpes simplex virus (HSV), Varicella zoster virus (VZV), and the use of pneumocystis prophylaxis. Conclusion: The low rates of vaccination indicate the need to improve physician and MM patient adherence and education regarding vaccination.

## 1. Introduction

New treatments introduced over the past decades have improved the survival of multiple myeloma (MM) patients [1,2,3]. Managing the complications of the disease and its treatments, including infections, is crucial as MM patients are living longer [4]. Infections remain a significant cause of morbidity and a leading cause of death in MM patients [5,6]. In a study of over 9000 MM patients, Blimark and colleagues observed that 22% of the deaths within the first year after diagnosis were from infections [7]. MM patients were 7 times more at risk for bacterial infections and 10 times more for viral infections compared to matched controls [7]. MM immunodeficiency is both humoral, with B-cell dysfunction leading to polyclonal hypogammaglobulinemia, and cellular, with T-cell, dendritic-cell, and NK-cell abnormalities [8]. Immunodeficiency starts as early as when the plasma cell disorder begins. Studies have shown an increased risk of infection in patients with monoclonal gammopathy of undetermined significance (MGUS) [9,10]. Low immune response to infections and vaccines has also been shown in MM patients and has been predictive of a higher risk of infection [11,12].

As a result of this immunosuppression, MM patients are notably at risk for developing infections involving encapsulated pathogens, such as *Streptococcus pneumoniae (SP*) and *Hemophilus influenzae b (Hib)*, as well as viral infections [13]. Despite the impaired response to vaccines [14], prophylactic vaccination has been recommended by the International Myeloma Working Group (IMWG) for influenza virus, *SP,* and *Hib* since 2002 [15]. These recommendations have since been re-emphasized in 2013 and 2017 [13,16].

To our knowledge, no previous population-based study has been conducted to evaluate the rate of vaccination and hospitalization imputable to infections in MM patients. Consequently, we tried to address these questions on a nationwide level using exhaustive data from the French national health insurance database. Therefore, the aims of our study were to assess *SP*, *Hib,* and influenza vaccination rates in newly diagnosed adult MM patients and to identify factors associated with a recommended vaccination.

## 2. Method

### 2.1. Data Sources

A retrospective pharmacoepidemiological cohort study was conducted using data from the French national health insurance database (SNDS). This database provides information on the healthcare coverage for approximately 98.8% of the French population [17]. The SNIIRAM (Système national d’information inter-régimes de l’Assurance maladie; National health insurance interplan information system) database contains exhaustive, anonymous individual data on patient healthcare reimbursements. It includes patient characteristics such as age, gender, vital status, and long-term and chronic diseases. It also includes data on ambulatory care with all reimbursed drugs from community pharmacies and all reimbursed medical interventions (with the French medical intervention classification coding (CCAM)). Through the French hospital discharge database (PMSI) several years of inpatient care all over France can be followed exhaustively. The PMSI includes the following data: the number of hospitalizations, admission and discharge dates, length of stay, type of hospital, and medical data coded according to the International Classification of Diseases, 10th revision (ICD-10) with diagnosis codes (main, related, and associated) [17,18].

### 2.2. Selection of MM Incident Cases

Patient selection is presented in the study flowchart (Figure 1). Data were extracted for patients presenting with a diagnosis of MM in the PMSI between 1 January 2010 and 31 December 2015. The ICD-10 code used was C90 “Multiple myeloma and malignant plasma cell neoplasms”.

Incident MM cases were identified according to the validated algorithm defined by Palmaro et al. [19]. This algorithm has a sensitivity of 90%, a specificity of 100%, and a predictive positive value of 60% [19]. The diagnosis date was then defined as the first hospitalization date for MM found in our dataset, according to the algorithm.

### 2.3. Observation and Study Periods

To ensure identification of new MM cases, we defined an observation period as the 12 months prior to the first date of MM diagnosis, which meant that we included only patients with the first diagnosis as of 1 January 2011 (Figure 1). 

The study period corresponds to the 24 months following the date of the diagnosis. Therefore, we included only patients with the first diagnosis up to 31 December 2013.

### 2.4. Definition of Outcomes

To meet the criteria for vaccination (primary outcome) patients had to have had at least one reimbursement for each of the three recommended vaccines during the study period. As influenza vaccine is recommended and free of charge for people 65 years of age and older in France [20], we searched for *SP* or *Hib* vaccines only for our secondary outcome. In fact, this is more MM specific since most of our study population was older than 65 years of age. Vaccines of interest were identified by their Anatomical Therapeutic Chemical (ATC) code: influenza vaccines (J07BB), *SP* vaccines (J07AL), and *Hib* vaccines (J07AG) [21].

Our analyses included incident MM cases with reimbursement of the three vaccines or at least an *SP* or *Hib* vaccine (depending on the outcome) during the study period. Patients with vaccine reimbursement were compared to incident MM cases without reimbursement of the three vaccines or of an *SP* or *Hib* vaccine (depending on the outcome) during the study period.

### 2.5. Covariates

The data source that we used comes from medicoadministrative data. In addition, some precise clinical data cannot be available (i.e., patient weight, proteinuria, etc.), while data that are difficult to collect in clinical practices are possible such as the level of consumption of care generating reimbursement. In addition, among the available data, we have chosen to select the data below.

We included the following covariates to compare vaccinated to nonvaccinated patients:-At diagnosis: age, gender, and complementary universal health insurance (CMU-C). In France, this supplementary insurance is available free of charge for people with a low income who are entitled to universal healthcare coverage.-During the observation period: Comorbidities were assessed by calculating a SNDS database adaptation of the Charlson Comorbidity Index [22,23,24]. We used the Charlson items and French recommendations for *SP* vaccination [25] to identify patients with a dual recommendation for *SP* vaccination (MM and another disease). We also included the healthcare consumption profile: number of different drugs used (categorized as ATC classes), number of different drugs used excluding vaccines (categorized as ATC classes), reimbursed vaccines (none versus at least one), number of medical visits (as a continuous variable), and number of hospital stays (none versus at least one). Lastly, we included two socioeconomic variables calculated using the community (smallest administrative unit in France) code [26]: the patient geographic area (urban versus rural) and the Fdep09, a deprivation index [27], with patients in the fifth quintile being the most deprived.-During the study period: antiviral prophylaxis (Herpes simplex virus (HSV) and Varicella-zoster-virus (VZV)) with at least two valaciclovir (ATC code J05AB11) reimbursements and *pneumocystis jirovecii* prophylaxis with at least two cotrimoxazole (ATC code J01EE01) or two pentamidine (ATC code P01CX01) reimbursements.

### 2.6. Analyses

Sociodemographic and medical characteristics of patients were described according to vaccinations (all three vaccines, *SP* or *Hib* vaccine, and no vaccine). Qualitative variables were expressed in frequencies and percentages and compared using the chi-squared test or Fisher’s exact test. Quantitative variables were expressed as mean and standard deviation (or median and interquartile range (IQR), if relevant) and associations were determined using the Student’s *t*-test or Wilcoxon test (if the variable was not normally distributed). All tests with a two-sided *p* value < 0.05 were considered significant.

For each of the vaccines considered, we recorded the date of dispensing as the date of vaccination. We then categorized vaccination dates in three periods: the first year after MM diagnosis, between the first and the second year after MM diagnosis, or between MM diagnosis and 2 years after, as there is no recommended vaccination period in the IMWG guidelines [13].

Factors associated with vaccination were examined using a logistic regression model with a backward stepwise elimination process. Age and gender were variables forced in the initial model. Factors associated with vaccination in bivariate analysis (*p* < 0.25) were included in the initial model (factors among: Charlson Comorbidity Index, *SP* vaccination recommended, Fdep99 deprivation index, complementary universal health insurance, geographic area, number of medical visits during observation, patients with at least one hospital stay during observation, number of nonvaccine drugs used during observation, patients vaccinated during observation, HSV-VZV prophylaxis during study, and *P. Jirovecii* prophylaxis during study). Potential multicollinearity was examined based on Besley’s criteria. The final model only retained statistically significant variables (*p* < 0.05). The goodness of fit of the final model was evaluated with the Hosmer–Lemeshow test and considered acceptable if the *p*-value was < 0.05.

All data analyses were carried out with SAS 9.4 software (SAS Institute, Cary, NC, USA).

## 3. Results

### 3.1. Characteristics of MM Patients

From 1 January 2010 to 31 December 2013, 36,990 subjects presented at least one MM ICD-10 code in France. After exclusion of prevalent cases, 22,831 MM cases were identified as incident (Figure 2). The median age was 74 years (IQR 64–82), with 11,797 (51.7%) male patients. During the 24-month study period, 3461 (15.2% of newly diagnosed patients) deaths occurred with 2352 (10.3% of newly diagnosed patients) deaths during the first year after MM diagnosis. Sociodemographic and medical characteristics, according to our primary outcome during the study period, are presented in Table 1. With respect to our secondary outcome, the same characteristics are presented in Table 2. 

### 3.2. Vaccine Use in MM Patients

Among the 22,831 newly diagnosed MM patients (Table 3), 6517 (28.5%) had at least one reimbursement for an influenza vaccine in year one after diagnosis and 5960 (26.1%) during year two after MM diagnosis. Regarding the *SP* vaccine, 1353 patients (5.9%) had at least one reimbursement during year one after MM diagnosis. All in all, during the 2 years after MM diagnosis, *SP* vaccination was initiated for 2350 (10.3%) patients. Moreover, 199 patients (0.9%) had an *Hib* vaccine reimbursement in year one after MM diagnosis and a total of 316 (1.4%) had an *Hib* vaccine reimbursement during the 2 years after MM diagnosis.

### 3.3. Factors Associated with Vaccination

Table 4 and Table 5 present the results of the bivariate and multivariate logistic regression models for the primary and secondary outcome, respectively.

Regarding our primary outcome, young age, vaccination during the observation period, and HSV-VZV or *P. Jirovecii* prophylaxis during the study period were associated with influenza, *SP*, and *Hib* vaccinations during the study period in the bivariate analyses. These associations persisted in the multivariate model.

Regarding our secondary outcome in the multivariate model, *SP* or *Hib* vaccination during the study period was associated with young age, male sex, an absence of comorbidities, an absence of medical consultations or hospitalizations, medication consumption and vaccinations during the observation period, and HSV-VZV or *P. Jirovecii* prophylaxis during the study period.

## 4. Discussion

Our study shows that the percentage of patients vaccinated for the 3 recommended vaccines is very low (0.7%). Patients vaccinated for *Streptococcus pneumoniae* (*SP*) or for *Hemophilus influenzae b* (*Hib*) represent only 10.4% of our sample. This first nationwide study indicates a low rate of MM patients vaccinated against influenza, *SP*, and *Hib* in France despite the publication of guidelines.

In the 2 years following MM diagnosis, only 0.7% of the patients had at least one reimbursement for all three recommended vaccines. Vaccination rates for influenza in year one and year two after MM diagnosis were 28.5% and 26.1%, respectively. During the 2 years following MM diagnosis, 10.3% of the patients had at least one *SP* vaccine reimbursement and 1.4% of the patients had at least one *Hib* vaccine reimbursement. 

One small retrospective American study with 411 MM patients from 2012 to 2014 found that 58% had received at least one *SP* vaccine [28]. To our knowledge, this is the only other study of vaccination rates in MM patients. Two other studies used the SNDS to assess vaccination rates in immunocompromised patients. The first involved 423 idiopathic thrombocytopenic purpura patients and showed that 32.4% of the patients were vaccinated against *SP* and 18.9% against *Hib* [29]. The second-known study included 101 patients with moderate to severe psoriasis treated with a biological drug and showed vaccination rates of 17.8% and 14.9% for influenza and *SP*, respectively [30]. Our vaccination rates are lower than in the previously cited articles. Considering that most of our patients were 70 years of age or older with comorbidities, we might have missed *SP* or *Hib* vaccination before MM diagnosis. Our influenza vaccination rates are also low (yearly rate of 26–29%) and therefore consistent throughout.

Overall, these results underscore that physician and patient adherence or education regarding vaccination is quite poor in France. Despite the fact that there is no clear data on the clinical impact of these vaccinations [31], it could be argued that this population is exposed to many disease and treatment-related risk factors of life-threatening infections and should be better protected. 

It is interesting to note that in this study, we also found that young age, an absence of comorbidity, a higher consumption of medications including vaccines, HSV-VZV and pneumocystis prophylaxis were associated with higher rates of vaccination. It indicates that oddly enough, patients with a lower risk of life-threatening infections are more likely to be vaccinated. In contrast, patients with comorbidities who often consult physicians and who have a history of hospitalizations are less likely to be vaccinated, despite more significant benefits of vaccination in this population. This might be due to the fact that better attention is paid to younger and healthier patients with a longer life expectancy. We can assume that people who are more health conscious are less likely to have comorbidity and are more likely to get vaccinated. These patient profiles may be more favorable to be vaccinated, or it is these patients who are more favorably offered to be vaccinated. 

Another explanation could be a measurement bias if older hospitalized patients were vaccinated during hospitalization as we could not account for in-hospital vaccination. However, this seems to be rare, because in France, vaccinations are entrusted to general practitioners [29]. There is also no guideline for in-hospital vaccination in France [29].

The patient’s geographic area of residence does not seem to be influenced by the analysis of the primary outcome. As the result is not significant, it is not possible to know if it is by an absence of effect or if it is by a lack of potency (152 patients among the 22,679 met the main endpoint). On the secondary objective, in univariate analysis, patients living in rural areas seem to be more likely to be vaccinated. Is it because of care in smaller health establishments or is it specifically associated with the place of residence? Further studies are needed to analyze this effect more precisely. Likewise, it would be very interesting to study whether there is heterogeneity between the different regions of French territory. Here too, further studies are necessary in order to be able to properly study this question.

Our study has some limitations that should be discussed. Because it was a study conducted using a health insurance database, identification of MM patients relied on ICD-10 codes. The possibility of miscoding cannot be fully excluded [32,33]. The algorithm we used to detect incident MM cases is also not flawless despite having a sensitivity and specificity of 90.4% and 99.7%, respectively [19]. Two factors might also have led to an underestimation of vaccine exposure. First, we could not obtain information regarding the 5 years prior to MM diagnosis. As a result, patients vaccinated before MM diagnosis might have been misclassified as nonvaccinated. However, *SP* and *Hib* vaccines are only recommended in France for infants under 18 months of age [34]. They are also recommended in case of risk factors for invasive *SP* or *Hib* infections such as immunodeficiency and chronic diseases [25]. There are concerns about the long-term effectiveness of these vaccines [35]. Consequently, revaccination is recommended when the previous dose of *SP* vaccine was administered more than 5 years earlier (10 years for *Hib*) [36]. We also might have underestimated vaccine exposure when patients had a chronic disease requiring *SP* vaccination in the 5 years prior to MM diagnosis. Considering that almost half of our patients had a condition requiring vaccination against *SP*, our results regarding this specific vaccine might have been altered. Second and as discussed previously, we only assessed vaccine exposure through hospital vaccine reimbursement. 

Lastly, the French policy for *SP* vaccination changed during the study period and now recommends a prime-boost strategy with a pneumococcal polysaccharide vaccine first, followed by a pneumococcal conjugate vaccine 8 weeks later [25]. The impact of this modification in clinical practice remains unknown. Therefore, large pharmacoepidemiological studies to assess the clinical impact of these vaccinations in this population of patients would be welcomed. 

## 5. Conclusions

All things considered, this study highlights insufficient rates of recommended vaccinations in French MM patients. More attention should be paid to appropriate vaccination, starting during smoldering MM as the highest antibody responses are obtained in patients vaccinated in the early stages of the disease (before initiation of chemotherapy and advanced hypogammaglobulinemia) [37]. Further assessments of this type in different countries would identify whether this problem is observed elsewhere. Studies evaluating the impact of measures to encourage the use of vaccination would make it possible to better target patient profiles for whom particular attention should be paid.

## Figures and Tables

**Figure 1 vaccines-08-00722-f001:**
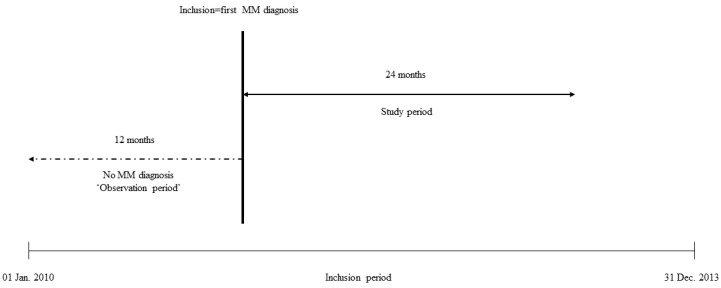
Study design—representation for a given patient.

**Figure 2 vaccines-08-00722-f002:**
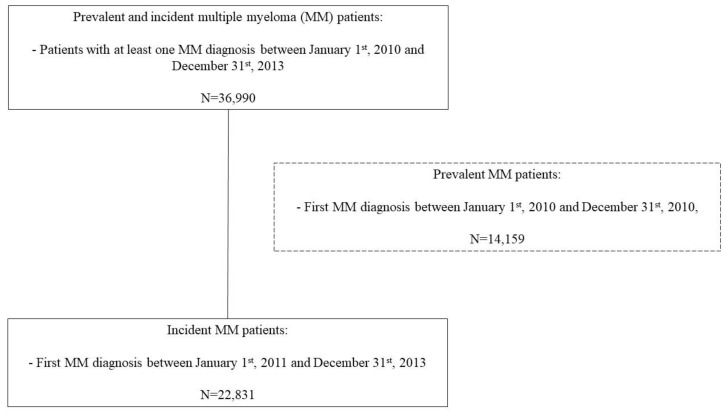
Flowchart of patients included in the study.

**Table 1 vaccines-08-00722-t001:** Sociodemographic and medical characteristics of patients during the observation period (12 months before multiple myeloma (MM) diagnosis)—primary outcome.

Characteristics	Total Population	Primary Outcome		*p*-Value
		Not All Recommended Vaccines	Influenza and *SP* and *Hib*	
Number of subjects, *n* (%)	22,831	22,679 (99.3)	152 (0.7)	
Age (years), median (IQR)	74 (64–82)	74 (64–82)	66 (62–75)	<0.0001
Females, *n* (%)	11,034 (48.3)	10,964 (48.3)	70 (46.1)	0.5731
Charlson Comorbidity Index, *n* (%)				0.0418
0	11,632 (50.9)	11,540 (50.9)	92 (60.5)	
1–2	6737 (29.5)	6696 (29.5)	41 (27.0)	
3–4	2119 (9.3)	2113 (9.3)	6 (3.9)	
≥5	2343 (10.3)	2330 (10.3)	13 (8.6)	
*SP* vaccination recommendation not related to MM, *n* (%)	11,099 (48.6)	11,028 (48.6)	71 (46.7)	0.6376
Fdep99 deprivation index, *n* (%)				0.4765
1st quintile	4213 (18.4)	4180 (18.4)	33 (21.6)	
2nd quintile	4067 (17.8)	4040 (17.8)	27 (17.8)	
3rd quintile	4145 (18.2)	4117 (18.1)	28 (18.4)	
4th quintile	4281 (18.8)	4254 (18.8)	27 (17.8)	
5th quintile	4264 (18.7)	4233 (18.7)	31 (20.4)	
Unknown	1861 (8.1)	1855 (8.2)	6 (4.0)	
Complementary universal health insurance (CMU-C), *n* (%)	1285 (5.6)	1279 (5.6)	6 (4.0)	0.3669
Geographic area, *n* (%)				0.9821
Urban	13,008 (57.0)	12,921 (57.0)	87 (57.2)	
Rural	4491 (19.7)	4462 (19.7)	29 (19.1)	
Unknown	5332 (23.3)	5296 (23.3)	36 (23.7)	
Number of medical visits, median (IQR)	13 (8–19)	13 (8–19)	12.5 (8–17)	0.1336
Patients with at least one hospital stay, *n* (%)	8579 (37.6)	8528 (37.6)	51 (33.6)	0.3041
Number of drugs used, excluding vaccines, median (IQR)	17 (11–24)	17 (11–24)	17 (12.5–23)	0.5273
Vaccinated patients, *n* (%)	9587 (42.0)	9946 (41.9)	91 (59.9)	<0.0001

**Table 2 vaccines-08-00722-t002:** Sociodemographic and medical characteristics of patients during the observation period (12 months before MM diagnosis)—secondary outcome.

Characteristics	Total Population	Secondary Outcome		*p*-Value
		No *SP* or *Hib*	*SP* or *Hib*	
Number of subjects, *n* (%)	22,831	20,466 (89.6)	2365 (10.4)	
Age (years), median (IQR)	74 (64–82)	75 (64–82)	67 (60–77)	<0.0001
Females, n (%)	11,034 (48.3)	9965 (48.7)	1069 (45.2)	0.0013
Charlson Comorbidity Index, *n* (%)				<0.0001
0	11,632 (50.9)	10,249 (50.1)	1383 (58.5)	
1–2	6737 (29.5)	6086 (29.7)	651 (27.5)	
3–4	2119 (9.3)	1984 (9.7)	135 (5.7)	
≥5	2343 (10.3)	2147 (10.5)	196 (8.3)	
*SP* vaccination recommendation not related to MM, *n* (%)	11,099 (48.6)	10,090 (49.3)	1009 (42.7)	<0.0001
Fdep99 deprivation index, *n* (%)				<0.0001
1st quintile	4213 (18.4)	3722 (18.2)	491 (20.8)	
2nd quintile	4067 (17.8)	3609 (17.6)	458 (19.4)	
3rd quintile	4145 (18.2)	3677 (18.0)	468 (19.8)	
4th quintile	4281 (18.8)	3832 (18.7)	449 (19.0)	
5th quintile	4264 (18.7)	3863 (18.9)	401 (16.9)	
Unknown	1861 (8.1)	1763 (8.6)	98 (4.1)	
Complementary universal health insurance, *n* (%)	1285 (5.6)	1158 (5.7)	127 (5.4)	0.5648
Geographic area, *n* (%)				0.0003
Urban	13,008 (57.0)	11,745 (57.4)	1263 (53.4)	
Rural	4491 (19.7)	4011 (19.6)	480 (20.3)	
Unknown	5332 (23.3)	4710 (23.0)	622 (26.3)	
Number of medical visits, median (IQR)	13 (8–19)	13 (8–19)	12.5 (8–18)	0.0025
Patients with at least one hospital stay, *n* (%)	8579 (37.6)	7811 (38.2)	768 (32.5)	<0.0001
Number of drugs used, median (IQR)	18 (12–25)	18 (12–24)	18 (12–25)	0.0015
Number of drugs used, excluding vaccines, median (IQR)	17 (11–24)	17 (11–24)	18 (12–24)	0.0017
Vaccinated patients, *n* (%)	9587 (42.0)	8550 (41.8)	1037 (43.9)	0.0533

**Table 3 vaccines-08-00722-t003:** Details of vaccinated patients for the study period (24 months after MM diagnosis).

Vaccinated Patients, *n* (%)	Time after MM Diagnosis		
	0–12 Months	12–24 Months	0–24 Months
Against influenza	6517 (28.5)	5960 (26.1)	8000 (35.1)
Against S.p.	1353 (5.9)	1149 (5.0)	2350 (10.3)
Against H.i.b.	199 (0.9)	125 (0.6)	316 (1.4)

S.p: *Streptococcus pneumoniae*; H.i.b.: *Hemophilus influenzae b*; MM: multiple myeloma.

**Table 4 vaccines-08-00722-t004:** Logistic regression model for factors associated with vaccination against influenza, *SP,* and *Hib* during the study period—primary outcome.

Characteristics	Crude OR (95% CI)	*p*-Value	Adjusted OR (95% CI)	*p*-Value
Age (year)	0.97 (0.96–0.98)	<0.0001	0.98 (0.97–0.99)	0.0132
Female gender	0.91 (0.66–1.26)	0.5732	1.00 (0.73–1.39)	0.9853
Charlson Comorbidity Index		0.0507	–	–
0	1			
1–2	0.77 (0.53–1.11)			
3–4	0.36 (0.16–0.82)			
≥5	0.70 (0.39–1.25)			
*SP* vaccination recommended	0.93 (0.67–1.28)	0.6377	–	–
Fdep99 deprivation index		0.5024	–	–
1st quintile	1			
2nd quintile	0.85 (0.51–1.41)			
3rd quintile	0.86 (0.52–1.43)			
4th quintile	0.80 (0.48–1.34)			
5th quintile	0.93 (0.57–1.52)			
Unknown	0.41 (0.17–0.98)			
Complementary universal health insurance	0.69 (0.30–1.56)	0.3697	–	–
Geographic area		0.9824	–	–
Urban	1			
Rural	0.97 (0.63–1.47)			
Unknown	1.01 (0.68–1.49)			
Number of medical visits during observation	0.99 (0.98–1.01)	0.4251	–	–
Patients with at least one hospital stay during observation	0.84 (0.60–1.18)	0.3048	–	–
Number of nonvaccine drugs used during observation	1.01 (0.99–1.02)	0.4004	–	–
Patients vaccinated during observation	2.07 (1.50–2.87)	<0.0001	3.00 (2.11–4.25)	<0.0001
HSV-VZV prophylaxis during study	5.87 (3.94–8.76)	<0.0001	3.15 (1.93–5.14)	<0.0001
*P. Jirovecii* prophylaxis during study	5.11 (3.67–7.11)	<0.0001	2.55 (1.70–3.80)	<0.0001

CI confidence interval, OR odds ratio. Logistic regression model adjusted for age, female gender, patients vaccinated during observation, HSV-VZV prophylaxis during study, and *P. Jirovecii* prophylaxis during study.

**Table 5 vaccines-08-00722-t005:** Logistic regression model for factors associated with vaccination against *SP* or *Hib* during the study period—secondary outcome.

Characteristics	Crude OR (95% CI)	*p* Value	Adjusted OR (95% CI)	*p* Value
Age (year)	0.970 (0.967–0.974)	<0.0001	0.98 (0.97–0.99)	<0.0001
Female gender	0.87 (0.80–0.95)	0.0013	0.91(0.83–0.99)	0.0324
Charlson Comorbidity Index		<0.0001		0.0448
0	1		1	
1–2	0.79 (0.72–0.88)		1.00 (0.90–1.11)	
3–4	0.50 (0.42–0.61)		0.83 (0.68–1.00)	
≥5	0.68 (0.58–0.79)		0.84 (0.71–0.98)	
*SP* vaccination recommended	0.77 (0.70–0.84)	<0.0001	–	–
Fdep99 deprivation index		<0.0001		<0.0001
1st quintile	1		1	
2nd quintile	0.96 (0.84–1.10)		0.97 (0.85–1.11)	
3rd quintile	0.97 (0.84–1.10)		1.03 (0.90–1.18)	
4th quintile	0.89 (0.78–1.02)		0.96 (0.84–1.11)	
5th quintile	0.79 (0.69–0.91)		0.87 (0.75–1.00)	
Unknown	0.42 (0.34–0.53)		0.54 (0.43–0.68)	
Complementary universal health insurance	0.95 (0.78–1.14)	0.5648	–	–
Geographic area		0.0003	–	–
Urban	1			
Rural	1.11 (1.00–1.24)			
Unknown	1.23 (1.11–1.36)			
Number of medical visits during observation	1.00 (0.99–1.00)	0.2455	0.99 (0.98–1.00)	0.0226
Patients with at least one hospital stay during observation	0.78 (0.71–0.85)	<0.0001	0.89 (0.81–0.99)	0.0256
Number of nonvaccine drugs used during observation	1.01 (1.00–1.01)	0.0003	1.01 (1.00–1.02)	0.0002
Patients vaccinated during observation	1.09 (1.00–1.19)	0.0534	1.39 (1.26–1.53)	<0.0001
HSV-VZV prophylaxis during study	3.01 (2.75–3.29)	<0.0001	2.11 (1.89–2.36)	<0.0001
*P. Jirovecii* prophylaxis during study	2.44 (2.23–2.66)	<0.0001	1.27 (1.14–1.41)	<0.0001

CI confidence interval, OR odds ratio. Logistic regression model adjusted for age, female gender, Charlson Comorbidity Index, Fdep99 deprivation index, number of medical visits during observation, patients with at least one hospital stay during observation, number of nonvaccine drugs used during observation, patients vaccinated during observation, HSV-VZV prophylaxis during study, and *P. Jirovecii* prophylaxis during study.

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
