# Peer review of "Vaccination during the First Diagnosis of Multiple Myeloma: A Cohort Study of the French National Health Insurance Database"

_vaccines, 2020, doi:10.3390/vaccines8040722_

Round 1

Reviewer 1 Report

This interesting study looked at the proportion of multiple myeloma patients who were vaccinated, and also the influencing factors of the vaccination pattern. Several concerns were raised when I read the manuscript.

  1. Lines 126-147: The authors have considered a whole bunch of covariates. It would be nice if they could provide more reasons behind the inclusion of these covariates;
  2. Table 1: apart from ‘no vaccines’ and ‘all vaccines’ groups, there are other groups (one vaccine and two vaccines). Would the authors please add two columns in this table and present the information on the individuals with one/two vaccines?
  3. There may be some reasons behind the findings that young age and an absence of comorbidity were associated with higher rates of vaccination. For example, people who are more health-conscious are less likely to have co-morbidity, and are more likely to get vaccinated. It would be great if the authors could discuss these findings in a more in-depth manner.

Author Response

First, we would like to thank the first referee for his/her review and wise comments. We will try to answer as clearly as possible to your comments and question about this article.

  1. Lines 126-147: The authors have considered a whole bunch of covariates. It would be nice if they could provide more reasons behind the inclusion of these covariates;
    • We thank the referee for his constructive remarks and we suggest to add the following sentences :
      • « The data source that we used comes from medico-administrative data. also, some precise clinical data cannot be available (i.e. patient weight, proteinuria, etc.) while data that are difficult to collect in clinical practices are possible such as the level of consumption of care generating reimbursement. Also, among the available data, we have chosen to select the data below »

  1. Table 1: apart from ‘no vaccines’ and ‘all vaccines’ groups, there are other groups (one vaccine and two vaccines). Would the authors please add two columns in this table and present the information on the individuals with one/two vaccines ?
    • We agree with the referee; the description of the different groups is not very easy to identify. In order not to overload Table 1 and to keep a clear description of the characteristics of the patients according to the main objective, we propose to add a new table 3, describing the distribution of the numbers according to the different vaccines.

Vaccinated patients, n (%)

Time after MM diagnosis

0-12 months

12-24 months

0-24 months

Against influenza

6517 (28.5)

5960 (26.1)

8000 (35,1)

Against S.p.

1352 (5.9)

1149 (5.0)

2347 (10.3)

Against H.i.b.

199 (0.9)

125 (0.6)

315 (1.4)

S.p: Streptococcus Pneumoniae; H.i.b.: Haemophilus Influenzae B; MM: Multiple Myeloma

Table 3. Details of vaccinated patients for the study period (24 months after MM diagnosis)

  1. There may be some reasons behind the findings that young age and an absence of comorbidity were associated with higher rates of vaccination. For example, people who are more health-conscious are less likely to have co-morbidity, and are more likely to get vaccinated. It would be great if the authors could discuss these findings in a more in-depth manner.
    • We thank the referee for his encouraging remarks and we fully agree with his proposals.; different factors can explain the results we observe. It is possible to hypothesize that among patients diagnosed with myeloma, a typical patient profile is more likely to be vaccinated, either because it is more favorable to vaccination, or because it is more visible. easily offered this treatment. We suggest adding the following paragraph to the discussion.
      • « This might be due to the fact, that better attention is paid to younger and healthier patients with a longer life expectancy. We can assume that people who are more health-conscious are less likely to have co-morbidity, and are more likely to get vaccinated. These patient profiles may be more favorable to be vaccinated or it is these patients who are more favorably offered to be vaccinated. »

Reviewer 2 Report

This is a well conducted study using a major database to determine vaccination of individuals during the first diagnosis of multiple myeloma. The results showed there was a need to improve patient adherance and education due to the low rates determined. The authors did recognise the limitiations of th study and where assumptions were made. However, the information gained is useful and should inform future practice and advice given to patients.

There are only minor points to be addressed:

  1. Page 4. Line 57. The 'also' needs removed from the sentence.
  2. Page 6. Line 113. This sentence is confusing. Do the authors mean 'at least one administration of each of the 3 vaccines during the study period was necessary to meet the criteria for vaccination'?

Author Response

First, we would like to thank the second referee for his/her review and wise comments. We will try to answer as clearly as possible to your comments and question about this article.

  1. Page 4. Line 57. The 'also' needs removed from the sentence.
  • We thank the referee for his careful reading and as proposed we have corrected in the text.

  1. Page 6. Line 113. This sentence is confusing. Do the authors mean 'at least one administration of each of the 3 vaccines during the study period was necessary to meet the criteria for vaccination'?
    • We confirm that to meet the primary vaccination endpoint, patients had to have had at least one reimbursement for each of the three recommended vaccines during the study period. We propose to rephrase the problematic sentence as follows:
      • “To meet the criteria for vaccination (primary outcome) patients had to have had at least one reimbursement for each of the three recommended vaccines during the study period.”

Reviewer 3 Report

This was a well designed and executed study of vaccination rates in MM patients. I have no concerns with the article as written, and offer only minor suggestions which the authors are free to accept or reject as they please.

I'd be interested to hear more about the covariates in the Discussion, particularly the differences observed between the primary and secondary outcome models. Why might geographic area be significant for the secondary outcome but not the primary? Etc.

Are there regional variations in vaccination rates? From a policy perspective, it might be interesting to view some of your findings mapped out, to see if physician/patient education is more needed in some areas than others for example.

Minor note - Is the line "considered acceptable if the p-value was >0.05" on line 167 a typo?

Author Response

First, we would like to thank the third referee for his/her review and wise comments. We will try to answer as clearly as possible to your comments and question about this article.

I'd be interested to hear more about the covariates in the Discussion, particularly the differences observed between the primary and secondary outcome models. Why might geographic area be significant for the secondary outcome but not the primary? Etc.

  • We thank the referee for his constructive remarks and we suggest to add the following sentences in the discussion section:
    • « The patient's geographic area of residence does not seem to be influenced by the analysis of the primary outcome. As the result is not significant, it is not possible to know if it is by an absence of effect or if it is by a lack of potency (152 patients among the 22,679 met the main endpoint). On the secondary objective, in univariate analysis, patients living in rural areas seem to be more likely to be vaccinated. Is it because of care in smaller health establishments or is it specifically associated with the place of residence? Further studies are needed to analyze this effect more precisely. »

Are there regional variations in vaccination rates? From a policy perspective, it might be interesting to view some of your findings mapped out, to see if physician/patient education is more needed in some areas than others for example.

  • - We thank the referee for his constructive remarks and we suggest to add the following sentences in the discussion section:
    • « Likewise, it would be very interesting to study whether there is heterogeneity between the different regions of French territory. Here too, further studies are necessary in order to be able to properly study this question. »

Minor note - Is the line "considered acceptable if the p-value was >0.05" on line 167 a typo?

  • - We thank the referee for his careful reading and as proposed we have corrected in the text as follows:
    • « The goodness of fit of the final model was evaluated with the Hosmer-Lemeshow test, and considered acceptable if the p-value was <0.05. »

Reviewer 4 Report

  1. SNIIRAM please specify the acronym
  2. lines 164-165 please add information on covariates used in the adjusted model.
  3. in tables please add covariates used for the adjusted model.
  4. why tables 3 and 4 did not show data for some variables? please add
  5. The discussion should better synthesize and describe the main results.
  6. the conclusion should be strengthened and more attention should be paid to the implications of these results.

Author Response

First, we would like to thank the forth referee for his/her review and wise comments. We will try to answer as clearly as possible to your comments and question about this article.

  1. SNIIRAM please specify the acronym
  • We thank the referee for his careful reading and as proposed we have corrected in the text as follows:
    • « SNIIRAM (Système national d'information inter-régimes de l'Assurance maladie; National health insurance inter-plan information system).

  1. lines 164-165 please add information on covariates used in the adjusted model.
    • We agree with the arbitrator and suggest modifying the paragraph as follows:
      • « Factors associated with vaccination were examined using a logistic regression model with a backward stepwise elimination process. Age and gender were variables forced in the initial model. Factors associated with vaccination in bivariate analysis (p<0.25) were included in the initial model (factors among : Charlson Comorbidity Index, SP vaccination recommended, Fdep99 deprivation index, Complementary universal health insurance, Geographic area, Number of medical visits during observation, Patients with at least one hospital stay during observation, Number of non-vaccine drugs used during observation, Patients vaccinated during observation, HSV-VZV prophylaxis during study and P. Jirovecii prophylaxis during study). »

  1. In tables please add covariates used for the adjusted model.
  • We agree with the referee and as suggested we suggest adding the following sentences at the bottom of the tables concerned:
    • Table 4 : CI confidence interval, OR odds ratio. Logistic regression model adjusted for Age, female gender, Patients vaccinated during observation, HSV-VZV prophylaxis during study and P. Jirovecii prophylaxis during study.
    • Table 5 : CI confidence interval, OR odds ratio. Logistic regression model adjusted for Age, female gender, Charlson Comorbidity Index, Fdep99 deprivation index, Number of medical visits during observation, Patients with at least one hospital stay during observation, Number of non-vaccine drugs used during observation, Patients vaccinated during observation, HSV-VZV prophylaxis during study and P. Jirovecii prophylaxis during study »

  1. Why tables 3 and 4 did not show data for some variables? please add
  • The univariate analysis column (crude OR) takes a reference value (noted 1) and compares it to the other variables. From these data, the Factors associated with vaccination in bivariate analysis (p <0.25) and gender were included in the initial model. Also, for the column in multivariate analysis (Adjusted OR) only the selected rows were used and filled in.

  1. The discussion should better synthesize and describe the main results
    • We agree with the referee and we suggest adding the following sentences:
      • « Our study shows that the percentage of patients vaccinated for the 3 recommended vaccines is very low (0.7%). Patients vaccinated for Streptococcus pneumoniae (SP) or for Haemophilus influenzae b (Hib) represent only 10.4% of our sample. »

  1. The conclusion should be strengthened and more attention should be paid to the implications of these results.
    • We thank the referee and we suggest adding the following sentences:
      • “Further assessments of this type in different countries would identify whether this problem is observed elsewhere. Studies evaluating the impact of measures to encourage the use of vaccination would make it possible to better target patient profiles for whom particular attention should be paid.”

Round 2

Reviewer 1 Report

Thanks to the authors for addressing my previous comments.